# Gran1: A Granulysin-Derived Peptide with Potent Activity against Intracellular *Mycobacterium* *tuberculosis*

**DOI:** 10.3390/ijms22168392

**Published:** 2021-08-04

**Authors:** Reiner Noschka, Fanny Wondany, Gönül Kizilsavas, Tanja Weil, Gilbert Weidinger, Paul Walther, Jens Michaelis, Steffen Stenger

**Affiliations:** 1Institute of Medical Microbiology and Hygiene, University Hospital Ulm, 89081 Ulm, Germany; reiner-martin.noschka@uni-ulm.de; 2Institute of Biophysics, Ulm University, 89081 Ulm, Germany; fanny.weiss@uni-ulm.de (F.W.); jens.michaelis@uni-ulm.de (J.M.); 3Max Planck Institute for Polymer Research, 55128 Mainz, Germany; kizilsavas@mpip-mainz.mpg.de (G.K.); weil@mpip-mainz.mpg.de (T.W.); 4Institute of Biochemistry and Molecular Biology, Ulm University, 89081 Ulm, Germany; gilbert.weidinger@uni-ulm.de; 5Central Facility for Electron Microscopy, Ulm University, 89081 Ulm, Germany; paul.walther@uni-ulm.de

**Keywords:** tuberculosis, antimicrobial peptide, Granulysin, human, macrophages

## Abstract

Granulysin is an antimicrobial peptide (AMP) expressed by human T-lymphocytes and natural killer cells. Despite a remarkably broad antimicrobial spectrum, its implementation into clinical practice has been hampered by its large size and off-target effects. To circumvent these limitations, we synthesized a 29 amino acid fragment within the putative cytolytic site of Granulysin (termed “Gran1”). We evaluated the antimicrobial activity of Gran1 against the major human pathogen *Mycobacterium tuberculosis* (*Mtb*) and a panel of clinically relevant non-tuberculous mycobacteria which are notoriously difficult to treat. Gran1 efficiently inhibited the mycobacterial proliferation in the low micro molar range. Super-resolution fluorescence microscopy and scanning electron microscopy indicated that Gran1 interacts with the surface of *Mtb*, causing lethal distortions of the cell wall. Importantly, Gran1 showed no off-target effects (cytokine release, chemotaxis, cell death) in primary human cells or zebrafish embryos (cytotoxicity, developmental toxicity, neurotoxicity, cardiotoxicity). Gran1 was selectively internalized by macrophages, the major host cell of *Mtb*, and restricted the proliferation of the pathogen. Our results demonstrate that the hypothesis-driven design of AMPs is a powerful approach for the identification of small bioactive compounds with specific antimicrobial activity. Gran1 is a promising component for the design of AMP-containing nanoparticles with selective activity and favorable pharmacokinetics to be pushed forward into experimental in vivo models of infectious diseases, most notably tuberculosis.

## 1. Introduction

Granulysin is an antimicrobial peptide (AMP) which is stored in the granules of human T-lymphocytes and natural killer cells [1]. The recombinant 9 kDa form of the protein is active against fungi, parasites, and bacteria, including *Mycobacterium tuberculosis* (*Mtb*) [2]. Killing of the microbes is associated with the ability to disrupt cell membranes and to facilitate the granzyme-mediated perturbation of mitochondria [2,3]. Granulysin also interacts with eukaryotic cell membranes, resulting in apoptosis of healthy and malignant cells [4,5]. As with several other AMPs, Granulysin has immune modulatory properties, and induces chemotaxis, the maturation of macrophages, and the release of pro-inflammatory cytokines [6,7]. While the combination of inflammation and apoptosis is beneficial for fighting malignant diseases, it is a double-edged sword in the treatment of chronic diseases, where (hyper)-inflammation is the pathophysiological hallmark of tissue destruction and severe disease. Accordingly, Granulysin mediates tissue damage in graft-versus-host disease and inflammatory skin diseases such as folliculitis, psoriasis, acne, or Steven Johnsons Syndrome [8]. Besides these off-target effects, the large size of Granulysin (9 kDa, 74 aa) accounts for rapid degradation in serum, the requirement of high micromolar concentrations for activity, and high costs for manufacturing. Finally, Granulysin alone is not active against intracellular pathogens, because it requires the support of perforin and granzymes to enter the intracellular compartment [2,9]. One approach to overcome these limitations is the identification of small Granulysin-derived peptides with specific bioactivities such as killing microbes, lysing cancer cells, or promoting inflammation [10,11,12,13,14]. Recent studies have demonstrated that helix 2 and helix 3 are essential for the antimicrobial activity of Granulysin [13,15]. By studying overlapping peptides, it was shown that the peptide “G8” (23–51 aa) lysis the extracellular, gram negative rod *Salmonella typhimurium* [13]. Based on these findings, we hypothesized that “G8” is active against extra- and intracellular *Mtb* without causing an inflammatory response. We selected mycobacteria as the target organisms because (i) the parental Granulysin is active against extracellular *Mtb* [2], (ii) mycobacteria are major human pathogens, (iii) mycobacteria are intrinsically resistant against most classes of antibiotics, (iv) the occurrence of infections with drug resistance strains is increasing (multidrug-resistant and extensively drug-resistant tuberculosis), [16] and (v) the available treatment which lasts at least 6 months and consists of a multidrug-regimen with potentially severe side effects. Therefore, novel therapeutic strategies against tuberculosis and related mycobacterial infections are desperately needed. Here, we demonstrate that Gran1 (previously described as “G8”) interacts with the mycobacterial cell wall and exerts antimicrobial activity against extra- and intracellular *Mtb* without triggering inflammatory immune responses or cell death in human cells or zebrafish embryos.

## 2. Results

### 2.1. Structure and Physicochemical Characteristics of Gran1

To investigate the interaction of Gran1 with extra- and intracellular mycobacteria, the fragment 23–51 aa within the cytolytic active site of the 9 kDa Granulysin was synthesized (Figure 1A). The peptide contains 29 amino acids, resulting in a molecular weight of 3.53 kDa. The ratio of hydrophobic residues within Gran1 is 34% and the net charge is +8, which are common characteristics for antimicrobial peptides and are predictive for an efficient interaction with the microbial cell wall. The Boman index, when used as a surrogate marker for the potential of an AMP to bind to membranes, is 5.18 kcal/mol, indicating high binding potential (Figure 1B). Structural analysis by circular dichroism (CD) measurements revealed that Gran1 follows a random coil-like structure at a neutral pH, which was maintained at acidic pH 5 and 6 (Figure 1C). The 3-dimensional structure of Gran1 in the solution (Figure 1D) was based on ^1^H Nuclear Resonance (NMR) spectroscopy data and calculations from Ambiguous Restraints for Iterative Assignment (ARIA) [17]. The 10 best fitting structure outputs of the calculations, based on NMR data (Figure 1D, left panel), and the state best fitting to the data that is in closest agreement with the Ramachandran plot (right panel), suggested that Gran1 does not form characteristic α-helices. Both CD and NMR spectroscopy indicated that Gran1 appears in a flexible, random coil-like conformation, in contrast to the helical structure of this fragment in the full-length protein. Overall, Gran1 fulfills the major physicochemical and structural criteria to interact with the mycobacterial cell wall.

### 2.2. Antibacterial Activity of Gran1 against Extracellular Mycobacteria

Based on these structural findings, we hypothesized that Gran1 inhibits the growth of bacterial pathogens. As model organisms, we initially evaluated the effect of Gran1 on non-tuberculous mycobacteria (NTM), which are becoming an increasing clinical problem and have an intrinsic resistance against the majority of conventional antibiotics. Gran1 showed dose-dependent activity against the three slow-growing mycobacterial species *M. kansasii*, *M. avium*, and *M. bovis BCG*, as determined by the incorporation of ^3^H-Uracil in the RNA of multiplying bacilli (Figure 2A). The inhibition of mycobacterial growth was already observed in the nanomolar range and reached a maximum of approx. 40% at higher concentrations. The overall activity was limited, which is typical for the effects of AMPs on mycobacteria [19].

In contrast, Gran1 showed no effect on the multiplication of the fast-growing, ubiquitous mycobacterial strains *M. fortuitum*, *M. vaccae*, or *M. smegmatis* (Figure 2B), which rarely causes infections in humans.

Next, we measured the antimicrobial activity of Gran1 on slow growing, highly pathogenic *Mtb*. Gran1 already limited the growth of *M. tuberculosis* by 15% at a concentration of 10 nM (Figure 3A). This effect peaked at a concentration of 1 µM, where growth was reduced by 38% when compared to un-treated control samples. Based on the dogma that AMP-mediated killing is mediated by the interaction of the positively charged peptide residues with the negatively charged bacterial cell walls [19], we investigated the interaction between Gran1 and extracellular *Mtb* with super-resolution fluorescence microscopy, using a STED microscope. Extracellular *Mtb* were labelled with a red fluorescent dye and were exposed to Gran1 (1 µM) for 30 min (Figure 3B). In un-treated samples (left panel), STED microscopy revealed rod-shape bacilli with a smooth cell surface. Gran1-treated bacteria (middle panel) showed an irregularly labelled surface, and the cell wall appeared fragmented, indicating alterations of the bacterial surface. To visualize the interaction of *Mtb* and Gran1, peptide-treated, labelled *Mtb* were stained with primary anti-Gran1 antibodies and a green-conjugated secondary antibody. In addition to the cell alterations, we observed a direct co-localization of the peptide and the mycobacterial surface (Figure 3B, right panel). To gain a detailed insight into the Gran1-mediated morphological effects on the *Mtb*-surface, we analyzed Gran1-exposed *Mtb* using scanning electron microscopy (Figure 3C). While untreated *Mtb* displayed a smooth surface with no alterations (Figure 3C, left panel), Gran-exposed *Mtb* showed surface lesions, including holes (middle panel), blebbing with irregular clusters, and distortions (right panel). We concluded that the mechanism of the Gran1-mediated growth inhibition of extracellular *Mtb* is related to the disruption of the integrity of the mycobacterial cell wall.

### 2.3. Off-Target Effects and Toxicity of Gran1

A prerequisite for the therapeutic application of AMPs is a favorable toxicity profile in the host organism. This includes limited induction of inflammatory off-target effects, lack of toxicity for eukaryotic cells, and no systemic developmental-, neuro- or cardiotoxicity. As markers for the induction of inflammation, we determined whether Gran1 induces cytokine release and/or chemotaxis in human macrophages and PBMC. The incubation of Gran1 in concentrations that limited mycobacterial growth (1 µM) failed to induce the secretion of IL-1β, IL-10, TNF-α, CCL-2, CCL-3, and CCL-5 in macrophages (Figure 4A) or PBMC (Figure 4B). Similarly, Gran1 did not promote the migration of freshly isolated PBMC in a chemotaxis chamber, as opposed to the secondary lymphoid-tissue chemokine SLC (mean migration factor 3.4), which was included as a positive control (Figure 4C). To evaluate the toxicity of Gran1 against primary human macrophages, the major host cell for *Mtb*, and hence the cellular target for antimycobacterial peptides, cells were incubated with Gran1, and metabolic activity was determined as a correlation indicating viability. Gran1 did not reduce the conversion of the substrate resazurin in macrophages obtained from five independent donors (Figure 4D) or in PBMC (data not shown). To test for toxicity in vivo, we used zebrafish embryos. We exposed embryos for 24 h, starting at 24 h post fertilization (hpf), when most organ systems have already developed and are functional. Transparency of the embryos allows for the evaluation, not only of mortality, but also of sublethal cytotoxicity (necrosis, lysis), developmental toxicity (developmental delay, malformations), or toxicity affecting specific organ systems, in particular cardiotoxicity (heart edema, reduced circulation) and neurotoxicity (reduced touch escape response), as mentioned in prior studies [20,21]. Gran1 neither effected embryo viability nor caused sublethal toxicity (Figure 4E). Taken together, concentrations of Gran1, which limit the growth of extracellular mycobacteria, show no functional off-target effects on immune cells and lack toxicity for primary human cells and live multicellular zebrafish embryos.

### 2.4. Internalization of Gran1 by Macrophages

Since *Mtb* is an intracellular pathogen, we next investigated whether Gran1 is preferentially internalized into macrophages, the major host cell of the pathogen. We co-incubated macrophages with autologous PBMC and treated the cultures with Gran1. Confocal laser microscopy revealed that Gran1 was efficiently internalized into MHC class II-positive macrophages (Figure 5A), whereas MHC class II negative cells (e.g., lymphocytes) did not show a signal. To quantify these results, we performed flow cytometry analysis and compared the intensity of Gran1-labelling in MHC class II-positive and -negative cells (Figure 5B, left and middle panel). The mean fluorescence intensity was five-fold higher in MHC class II-positive cells, showing that Gran1 is preferentially taken up by macrophages in mixed cell cultures with PBMC (Figure 5C, right panel). Uptake of Gran1 was markedly reduced when the cultures were incubated at 4 °C, indicating that this is an energy-dependent process, most likely phagocytosis (Figure 5C).

To exert its antimicrobial function, Gran1 needs to be taken up by *Mtb*-infected macrophages. Therefore, we investigated whether *Mtb* infection interferes with the uptake of Gran1 by exposing the *Mtb*-infected macrophages to Gran1 and analyzing cells by confocal laser scanning microscopy. Labelled Gran1 was detected in the majority (>95%) of macrophages independently of *Mtb*-infection (Figure 6). Gran1 was homogenously distributed throughout the cytoplasm and no specific enrichment in the vicinity of the bacilli was detected.

### 2.5. Antibacterial Activity of Gran1 against Intracellular Mtb

Since Gran1 kills extracellular mycobacteria and enters *Mtb*-infected macrophages, we addressed the key question of whether Gran1 inhibits the proliferation of *Mtb* residing inside primary human macrophages. Macrophages were infected with *Mtb* and incubated with Gran1. After 4 days, the number of live bacilli was determined by quantifying the number of CFU. The *n*-fold growth was normalized to the control cultures (broth) and revealed that the diluent alone had no effect on *Mtb* multiplication (Figure 7). In contrast, the mean *n*-fold growth of *Mtb* in Gran1-treated cultures was 0.64, as compared to the broth control, which demonstrated a statistically significant growth reduction of 36%.

Taken together, Gran1 inhibits the growth of extra- and intracellular mycobacteria in concentrations that are well tolerated in vitro (macrophages) and in vivo (zebrafish embryos), and do not trigger inflammatory off-target effects in macrophages or PBMC. Studies that are aimed at optimizing the pharmacokinetics of Gran1 to allow for efficacy studies in preclinical and ultimately clinical studies in mycobacterial infection are currently ongoing.

## 3. Discussion

The clinical application of large AMPs such as Granulysin is limited by off-target effects, poor penetration into the cytoplasm of microbial host cells, rapid degradation in human serum, and high manufacturing costs. Therefore, we investigated whether the specific cytolytic region of Granulysin (23–51 aa; Gran1) exerts antimicrobial activity against extra- and intracellular mycobacteria without inducing inflammatory immune responses. Our results demonstrated that the 3.53 kDa Gran1 kills slow-growing non-tuberculous mycobacteria and virulent *Mtb* inside human macrophages without triggering inflammatory immune responses such as cytokine release or chemotaxis. Thus, Gran1 is a promising candidate to be further evaluated for efficacy against microbial infection.

Prior studies showed that a conserved secondary structure of AMPs is crucial to maintain its biological activity [22,23]. Hence, we investigated the secondary structure of Gran1. We identified a random coil-like conformation without distinct alpha helices. The multiple arginines present in the peptide sequence, the two cysteines both embedded in the same tripeptide code VCR, and the predominance of amino acids with similar fingerprints (Ser-Cys, Asn-Asp-Phe-Trp) in the TOCSY spectrum, plus broad overlapping cross-peaks in the backbone HN-Hα coupling region, could interfere with the unequivocal signal assignment of the 29 amino acids. Additionally, heteronuclear measurements such as ^1^H-^13^C Heteronuclear Single Quantum Coherence (HSQC) and Heteronuclear Multiple Bond Correlation (HMBC) were unsuccessful due to a low concentration of the peptide, as related to the natural abundance of ^13^C. This problem will be addressed in future studies by using recombinant, rather than synthetic, Gran1 to allow for a precise analysis of the solution structure. We conclude, therefore, that even though the secondary structure of Gran1 differs from Granulysin, the antimicrobial activity of Gran1 was retained, in agreement with studies that have demonstrated the functional significance of helix2-loop2-helix3 in antimicrobial activity [15]. Furthermore, the four basic arginine residues which are required for membrane binding are conserved in the sequence of Gran1 sequence [7]. We hypothesized that the amino acid sequence is more critical for the antimicrobial function than the secondary structure, since Gran1 resembles the unmodified cytolytic active region (23–51 aa). Granulysin-derived peptides are active against extracellular *Staphylococcus aureus*, Streptococci [15], *Propionibacterium acnes* [10], *Vibrio cholera* [12], *Escherichia coli* [22], and *Mtb* [14,22]. Importantly two of these peptides had a protective effect against *Vibrio cholera* in mice [12], and the local application of a peptide-containing formulation improved the clinical presentation of severe acne in humans [11]. Here, we demonstrated that a Granulysin-derived peptide is active against a panel of mycobacterial species, including virulent *Mtb*. We used mycobacteria as target organisms because they cause severe disease in humans and are notoriously difficult to treat. Ongoing studies will demonstrate whether Gran1 is also active against extracellular, fast growing bacteria such as *Staphylococcus aureus*, *Klebsiella pneumoniae*, or *Pseudomonas aeruginosa*. Our results indicate that Gran1 has a selective activity against the slow-growing species *M. tuberculosis*, *M. kansasii*, *M. avium*, and the vaccine strain *M. bovis BCG*, but not the rapid-growing strains *M. fortuitum*, *M. vaccae*, and *M. smegmatis*. One possible explanation for this is that the effect of the AMP on the bacteria is slower than the growth rate of non-tuberculous mycobacteria (NTM) [24], so any harmful effect could be compensated by the rapid multiplication. Alternatively, the differential effect could be related to the selective activity of anti-tuberculosis drugs to fast- and slow-growing NTM [25,26]. For example, rifampin is less active against rapid-growing mycobacteria, based on a more efficient efflux activity, as compared to slow-growing mycobacteria [25]. By using super-resolution and electron microscopy studies, we found that Gran1 directly interacts with the mycobacterial cell wall, causing pore formation and distortions reminiscent of early studies performed with the parental Granulysin [2]. We hypothesize that, via the positively charged arginine residues, Gran1 interacts with the negatively charged mycobacterial cell wall, leading to deleterious disruption of cell wall integrity and consecutive disturbance of the ionic balance. Whether Gran1 interferes with specific molecular targets in the mycobacterial cell wall remains to be determined. STED microscopy, recently established for studying peptide/mycobacteria interaction ([27], Figure 3) provides a powerful tool with which to address these mechanistic questions.

We have demonstrated for the first time that a Granulysin-derived peptide can enter human macrophages to kill an intracellular pathogen. Notably, concentrations of Gran1 that induced growth inhibition did not induce apoptosis or necrosis in human macrophages but accumulated in the cytoplasm of un-infected and *Mtb*-infected macrophages (Figure 6). This distinguishes Gran1 from other antimicrobial peptides (e.g., Melittin), which preferably incorporates into cell membranes [28], most likely due to a significantly lower Boman index (0.57 kcal/mol; APD3 [18]). Even though we found distinct areas of co-localization in Gran1 and intracellular *Mtb* by confocal laser microscopy (Figure 6), the resolution of this technique does not allow for definitive conclusions, especially because of the homogenous distribution of the peptide throughout the cytoplasm. Gran1 might interact with the mycobacterial cell wall, as observed for extracellular *Mtb* (Figure 2). However, since *Mtb* resides within phagolysosomes and not in the cytoplasm, Gran1 may not gain access to the mycobacteria. In this case, Gran1 could modulate macrophage function and indirectly contribute to the growth inhibition of *Mtb*. Previously it was shown that Gran1/G8 activates sodium channels in human red blood cells, resulting in increased levels of intracellular sodium, chloride, and calcium ions and reduced levels of potassium [29]. This effect on the ion-microenvironment could also be active in macrophages and modulate pathways involved in the antimicrobial function of macrophages [30], for example the ATP-induced NLRP3 inflammasome activation [31]. Whether Gran1 directly affects antimycobacterial effector pathways, such as the vitamin D-cathelicidin axis [32,33], the fusion of phagosomes and lysosomes [34] or autophagy [35] is currently under investigation. In addition, we are currently fine-tuning the use of super-resolution microscopy to allow for a more definitive insight into the intracellular (co)-localization of Gran1 and mycobacteria.

One caveat when using large peptides for the treatment of infections is the risk of off-target effects not related to the intended antimicrobial activity. Several AMPs combine antimicrobial and immune-modulatory activities, most notably cathelicidin [36] and Granulysin [7,8], both of which are active against mycobacteria [2,37,38]. Specifically, Granulysin induces apoptosis in eukaryotic cells, is a chemoattractant, and triggers the release of immune modulatory cytokines (IL-10, IL-1β, IL-6 and IFN-α) [8]. While this immune modulatory effect can be a helpful bystander effect to combat acute infections, it may be harmful in diseases where inflammation is a major component of tissue destruction. Depending on the pathophysiology of the disease peptides, which avoid or suppress immune modulatory functions, may be beneficial. For example, a Granulysin-derived peptide was identified (31–50 aa) which combines antimicrobial activity against the causative bacterium *Propionibacterium acnes* with anti-inflammatory effects that contribute to the progression of disease [10]. In tuberculosis, especially in the later stages of disease, when the majority of the bacteria have been eliminated, inflammation contributes to tissue damage, most notably the destruction of lung parenchyma [39]. Under these circumstances, Gran1 might be particularly useful, because it maintains the antimicrobial activity of the parental peptide without inducing apoptosis, chemotaxis, or the release of immune modulatory cytokines.

Taken together, Gran1 is an AMP with antimicrobial activity against clinically relevant, slow-growing mycobacterial species, including virulent *Mtb*, without promoting inflammatory immune responses. In contrast to the parental protein Granulysin, it does not require the support of granzymes and perforin to inhibit the growth of intracellular *Mtb*. Our results pave the way for designing Gran1-containing nanoparticles, such as liposomes or mesoporous nanoparticles, which combine AMPs with conventional anti-tuberculosis compounds. These nanoparticles will serve to optimize the stability, pharmacokinetics, and biological activity of Gran1, which can then be evaluated as a treatment against mycobacterial infections in preclinical and ultimately clinical studies.

## 4. Materials and Methods

### 4.1. Peptide Synthesis

Gran1 (QRSVSNAATRVCRTGRSTWRDVCRNFMRR) was synthesized by PSL Heidelberg (PSL Heidelberg, Heidelberg, Germany) using F-moc chemistry. For visualization of Gran1, the peptide was conjugated N-terminally to the fluorescent dye Atto647N (PSL Heidelberg). Peptides were purified to >95% homogeneity by reverse-phase HPLC. Composition of each peptide was confirmed by amino acid analysis and mass spectrometry [40]. Stock peptide solutions (10 mg/mL) were prepared in Ampuwa (Fresenius Kabi, Bad Homburg, Germany)—aqua ad injectabilia. Physicochemical properties of Gran1 were predicted using the Antimicrobial Peptide Databank server APD3 [18] for amino acid composition, molecular weight, hydrophobic ratio, and Boman index.

### 4.2. Structural Analysis of Gran1

#### 4.2.1. Circular Dichroism Spectrometry

CD spectra were recorded on a JASCO J-1500 spectrometer (Jasco, Pfungstadt, Germany) in a 1 mm High Precision Cell by HellmaAnalytics (Müllheim, Germany). The samples with Gran1 were prepared at a concentration of 200 μM in 10 mM phosphate buffer, separately at pH 5, pH 6, and pH 7. Measurements were conducted using the following parameters: path length: 0.1 mm, scan rate: 5 nm/min, scan range: 260 nm–180 nm, data pitch: 0.2 nm, and data integration time: 2 sec. Data were processed in Spectra Analysis by JASCO (Jasco, Pfungstadt, Germany) and Excel (Microsoft Cooperation, Redmond, WA, USA).

#### 4.2.2. Nuclear Magnetic Resonance

For all 2D NMR experiments, 5 mg of Gran1 was dissolved in 450 mL aqua ad injectabilia (Ampuwa) and 50 mL D_2_O. All experiments were recorded on a BRUKER spectrometer (Billerica, MA, USA) operating at 850 MHz ^1^H frequency. Experiments were carried out at 298 K. Nuclear Overhauser Effect Spectroscopy (NOESY) spectra acquiring 2D homonuclear correlation via dipolar coupling with water suppression using watergate W5 pulse sequence with gradients [41] were recorded for a mixing time of 150, 300, and 450 ms, using 2 × 16k × 256 data matrices, corresponding to acquisition times of ~480 and 8 ms in the t1 and t2 dimensions, respectively. Next, 64 scans were acquired per t1 value. Through-bond connectivity was obtained from a Total Correlation Spectroscopy (TOCSY) spectrum, recorded with the MLEV-17 mixing scheme [42], with water suppression using 3-9-19 pulse sequence, and with gradients [43,44] using a 13 µs 90° pulse and for an 80 ms mixing period.

NMRFAM-Sparky was used for signal assignment and NOE signal volume determination [45]. For NOE signal integration, a gaussian fit was used, allowing peak motion and the adjusting of linewidths and baseline fitting. For the 3D structure calculation of Gran1, the software package Ambiguous Restraints for Iterative Assignment (ARIA) was used [17].

### 4.3. Source and Culture of Mycobacteria

The following mycobacterial strains were used (Table 1):

Mycobacteria were amplified, stored, and cultured as described previously [46]. Representative vials were thawed and enumerated for viable colony forming units (CFU) on Middlebrook 7H11 plates (BD Biosciences, Franklin Lakes, NJ, USA). Live-dead staining (BacLight, Invitrogen, Carlsbad, CA, USA) of bacterial suspensions with fluorochromic substrates revealed a viability of the bacteria >90% [47]. Prior to use, thawed aliquots were sonicated in a water bath for 10 min at 40 kHz and 110 W at room temperature to disrupt small aggregates of bacteria [48].

### 4.4. Antibodies and Reagents

The anti-Gran1 antibody was generated in cooperation with ImmunoGlobe (Himmelstadt, Germany). Rabbits were immunized by injecting intradermally synthetic Gran1 and the adjuvant Montanide for three times over a timespan of 12 weeks. Afterwards, the IgG fraction was isolated from serum using an IgG column. Purified anti-Gran1 antibody was stored at 4 °C in Tris-buffer with 0.02% NaN3. Cy2-conjugated donkey-anti-rabbit was acquired from Dianova (Hamburg, Germany) and Cy5-conjugated goat anti-mouse from Jackson ImmunoResearch (Westgrove, PA, USA). 4′,6-diamidino-2-phenylindole (DAPI), Atto594-conjugated goat anti-rabbit, and succinimidylester Atto647N were all purchased from Sigma–Aldrich (Steinheim, Germany). Succinimidylester AlexaFluor 647 and FITC-conjugated HLA-DR antibody were purchased from Invitrogen (Carlsbad, CA, USA), while unconjugated HLA-DR antibody was purchased from Leinco Technologies (St. Louis, MO, USA). Macrophage serum-free media (M-SFM) and AIM V serum-free media were purchased from Gibco, Thermo Fisher (Waltham, MA, USA). Middlebrook 7H9 broth was purchased from Becton Dickinson (BD Biosciences, Franklin lakes, NJ, USA). Staining buffer was prepared in PBS containing 0.5% Tween 80, 0.2 M sodium bicarbonate (both Roth, Karlsruhe, Germany) and pH was set to 8.8. FACS buffer contained 1% FCS (Biochrom, Berlin, Germany) and 0.1% sodium azide (VWR, Radnor, PA, USA) in PBS (Gibco, Carlsbad, CA, USA).

### 4.5. Chemotaxis Assay

Migration of PBMC was examined using a 96-well microchemotaxis chamber (NeuroProbe, Gaithersburg, MD, USA). Cells were re-suspended in AIM V medium. Different concentrations of Gran1 were placed in the lower compartment of the chamber and 0.1 × 10^6^ cells were added to the upper compartment in a total volume of 50 µL. Cells were allowed to migrate at 37 °C for 3 h. Filters were then washed with PBS and removed to transfer migrated cells from the lower compartment into FACS tubes. For quantification, 5 µL of fluorescent-red latex beads were added to each sample. Cells were acquired using a FACSCalibur flow cytometer (BD Biosciences, Franklin lakes, NJ, USA) and terminated when 5 × 10^4^ events in the latex beads gate were detected. The chemotaxis factor was calculated by dividing the number of migrated cells to Gran1 by the number of migrated cells to medium control.

### 4.6. Cytokine and Chemokine Release

For determination of cytokine and chemokine release 2 × 10^6^ human PBMC or 0.5 × 10^6^, macrophages were seeded in a 24-well plate and incubated overnight with Gran1 at 37 °C. LPS (100 ng/mL) or PHA (20µg/mL, for CCL-5 release) served as positive controls. Supernatants were collected and analyzed for TNF-α, IL-1β (Endogen, Waltham, MA, USA), IL-10, CCL-2, CCL-3, and CCL-5 (all R&D systems, Minneapolis, MN, USA) by ELISA, as suggested by the manufacturer. Absorption was measured at 450 nm using Infinite 200 Pro (Tecan, Männedorf, Switzerland) plate reader. Background signals were deducted for indicated ELISA specifically mentioned in the figure legend.

### 4.7. Growth of Extracellular Mycobacteria: ^3^H-Uracil Proliferation Assay

The activity of Gran1 against virulent *Mtb*, as well as non-tuberculous mycobacteria, was determined by measurement of RNA synthesis after the incorporation of radioactively-labelled 5.6-^3^H-Uracil (ART-0282, Biotrend, Cologne, Germany). Next, 2 × 10^6^ sonicated mycobacteria (virulent H37Rv or non-tuberculous mycobacteria, respectively) were incubated with Gran1 in middlebrook 7H9 broth in a 96-well plate. All samples were set up in triplicate, using 2 µg/mL rifampicin or 2 µg/mL clarithromycin for rapid-growing strains as a control, respectively. ^3^H-Uracil (0.3 μCi/well) was added after 72 h for fast-growing mycobacteria or 168 h for slow-growing mycobacteria, and cultures were incubated for additional 18 h. *Mtb* were then inactivated by treatment with 4% paraformaldehyde (PFA, Sigma–Aldrich, Steinheim, Germany) for 30 min and transferred onto glass fiber filters (Printed Filtermat A, PerkinElmer, Waltham, MA, USA) using a 96-well based Filtermat Harvester (Inotech, Nabburg, Germany). Fiber filters were dried in a microwave at 240 W for 5 min and sealed at 75 °C with a sheet of solid scintillant wax (MeltiLex, PerkinElmer, Waltham, MA, USA). Radioactivity was measured using a β-Counter (Sense Beta, Hidex, Turku, Finland). Antimicrobial activity (%) was calculated as counts per minute (cpm) of the treated sample/cpm of the un-treated sample × 100.

### 4.8. Stimulated Emission Depletion (STED) Microscopy

First, mycobacteria were fluorescently labelled as previously described [26]. Briefly, mycobacteria were washed with staining buffer, followed by incubation with succinimidyl ester Atto647N (1 µg/mL) for 1 h at 37 °C. Afterwards, bacteria were washed and resuspended in M-SFM. For investigating the interaction between Gran1 and *Mtb*, stained bacteria were seeded in a 12-well plate (Sarstedt, Nümbrecht, Germany) containing a sterile precision glass coverslip (170 ± 5 µm thickness, Carl Roth, Karlsruhe, Germany) coated with poly-L-lysine (Sigma–Aldrich, Steinheim, Germany). Then, unconjugated Gran1 (1 µM) was added and incubated for 30 min. Excess peptide was removed by rinsing the wells carefully. Coverslips were fixed with 4% PFA and washed with PBS. Bacteria were blocked and permeabilized for 2 h in 3% BSA and 0.3% TritonX-100 in PBS. This was followed by incubation overnight at 4 °C with primary anti-Gran1 antibody (1:1000). Afterwards, bacteria were incubated with secondary Atto594-conjugated goat anti-rabbit antibody (1:1000) for one hour at room temperature. Coverslips were mounted with Mowiol (Sigma–Aldrich, Steinheim, Germany) onto a SuperFrost plus microscope slides (R. Langenbrinck GmbH, Emmendingen, Germany) for analysis by STED microscopy. Images were captured using a home-built dual-color STED microscope described elsewhere [49]. Typically, an average power of ~0.8µW for each excitation beam (568 nm and 633 nm, respectively) and ~1.3 mW for each depletion beam (710 nm and 750 nm, respectively) was used. STED images were captured at a pixel size of 10 nm and a dwell time of 300 µsec, with a typical peak photon number of ~100 counts. Images were analyzed in ImageJ 1.53c. For better visualization, a Gaussian blur of σ = 1 and an intensity threshold of >5 counts were applied in each channel.

### 4.9. Scanning Electron Microscopy

First, 5 × 10^6^ *M. tuberculosis* were seeded in a 24-well plate in middlebrook 7H9 broth, followed by incubation with 1 µM Gran1 for 72 h. After infection, bacteria were harvested, transferred to screw caps, and centrifuged at 10.000 rpm for 10 min. Supernatant was discarded and the pellet of *Mtb* was resuspended and fixed in 100 µL 4% PFA for 20 min. Afterwards, bacteria were chemically fixed with 2.5% glutaraldehyde (in PBS and 1% saccharose) for one hour. Then, *Mtb* were post-fixed with OsO_4_ (2% in PBS) for one hour at room temperature to dehydrate samples gradually in 30, 50, 70, 90 and 100% propanol (5 min at each step) [50]. Bacteria were then critical point dried using carbon dioxide as translation medium (Critical Point Dryer CPD 030, Bal-Tec, Principality of Liechtenstein). Samples were rotary coated in a BAF 300 freeze etching device (Bal-Tec, Principality of Liechtenstein) by electron beam evaporation with 3 nm of platinum–carbon from an angle of 45°. Images were acquired using a Hitachi S-5200 in-lens field emission SEM (Hitachi High-Tech, Tokyo, Japan) at an accelerating voltage of 10 kV by using the secondary electron signal.

### 4.10. Confocal Laser Scanning Microscopy

For investigation of Gran1 cell-specific internalization, a mixed culture of primary human macrophages and autologous PBMC (ratio 1:1, total cell counts 0.1 × 10^6^) were seeded in 200 µL M-SFM in an 8-chamber slide (Thermo Fisher, Waltham, MA, USA). Followed by incubation with Gran1 for 2 h, cells were then fixed with 4% PFA and permeabilized for 10 min in 0.5% bovine serum albumin (BSA), 0.1% Triton X-100, and 0.05% Tween 20 (all Sigma–Aldrich, Steinheim, Germany) in phosphate buffered saline (PBS). Samples were blocked for one hour with blocking buffer (1% BSA, 0.1% Triton X-100 in PBS) and incubated at room temperature for one hour with primary anti-Gran1 (1:250) or MHC class II (1:300). Afterwards, cells were incubated with the secondary antibody Cy2-conjugated donkey anti-rabbit (1:200) or Cy5-conjugated goat anti-mouse (1:250), respectively, for one hour at room temperature. Cell nuclei were stained with 1 µg/mL DAPI for 10 min and slides were mounted with Aquatex (Merck, Darmstadt, Germany). Images were acquired using the inverted laser scanning confocal microscope LSM 710 (Zeiss, Oberkochen, Germany) and analyzed in ImageJ 1.53c. For investigation of Gran1-localization in infected macrophages, *Mtb* were stained with the fluorescein succinimidyl ester AlexaFluor 647 (1 mg/mL) as described [46]. Afterwards, macrophages were infected with stained *Mtb* at a theoretical multiplicity of infection of 50 for 16 h. Followed by incubation with Gran1 for 2 h, cells were then fixed and stained as mentioned above.

### 4.11. Toxicity of Gran1 against Macrophages and Zebrafish

For in vitro studies, 1 × 10^5^ macrophages were incubated with Gran1 for 24 h in a 96-well plate, followed by an addition of 10% PrestoBlue (Thermo Fisher, Waltham, MA, USA) for 20 min. Reduction of the non-fluorescent resazurin-based PrestoBlue to fluorescent resorufin by mitochondrial enzymes of viable cells allowed for the calculation of cytotoxicity as described [48].

For in vivo studies, wild-type zebrafish embryos (*Danio rerio*) were dechorionated at 24 h post-fertilization (hpf) using digestion with 1 mg/mL pronase (Sigma-Aldrich, Steinheim, Germany) in E3 medium (83 μM NaCl, 2.8 μM KCl, 5.5 μM 202 CaCl_2_, 5.5 μM MgSO_4_). In a 96-well plate, 3 embryos per well were exposed for 24 h to 100 µL of E3 containing Gran1 (1 µM). Two independent assays were performed, each with 10 × 3 embryos. The peptide solvent (Ampuwa, aqua ad injetibilia), diluted in E3, was used as negative control at the same amount as introduced by the peptide stock. As positive control for acute toxicity/cytotoxicity, the pleurocidin antimicrobial peptide NRC-03 (GRRKRKWLRRIGKGVKIIGGAALDHL-NH2) was used at a concentration of 6 µM, as described [20]. Abamectin at a concentration of 3.125 µM was used as positive control for neurotoxicity [51]. At 48 hpf (after 24 h of incubation), embryos were scored in a stereomicroscope for signs of acute toxicity/cytotoxicity (lysis and/or necrosis), developmental toxicity (delay and/or malformations), or cardiotoxicity (heart edema and/or reduced or absent circulation). Each embryo was also touched with a needle and the reduced or absent touch response (escape movements) was evaluated for signs of neurotoxicity if, and only if, no signs of acute toxicity were present in the same embryo. Embryos were categorized within each of these toxicity categories into several classes of severity. Chi-square test was used to calculate whether the distribution of embryos into toxicity classes [21] differed significantly between the negative control and the test substances.

### 4.12. Intracellular Detection of Gran1 by Flow Cytometry

For investigation of Gran1 cell-specific uptake, macrophages were seeded in sterile FACS tubes alongside autologous PBMC in a 1:1 ratio for a total cell count of 0.5 × 10^6^ cells in AIM V medium. Cells were incubated with Gran1-Atto647N for 2 h at 37 °C. Afterwards, cells were washed with FACS buffer and centrifuged for 10 min at 1300 rpm. Supernatant was discarded and cells were stained against MHC class II by a FITC-conjugated HLA-DR antibody (1:200). Sample analysis was performed using a FACSCalibur flow cytometer (BD Biosciences, Franklin Lakes, NJ, USA). Data analysis was performed using FlowJo Version 10.5.3 (BD Biosciences, Franklin Lakes, NJ, USA) and GraphPad Prism Version 8.2.1 (GraphPad Software, La Jolla, CA, USA).

### 4.13. Quantification of Intracellular Mycobacterial Growth

Human peripheral blood mononuclear cells (PBMC) were isolated from buffy coats of anonymous donors (Institute of Transfusion Medicine, Ulm University) by density gradient centrifugation (Ficoll-Paque Plus, GE Healthcare, Buckinghamshire, UK). Monocytes were selected from plastic adherence and thorough washing. For generation of monocyte-derived macrophages, cells were cultured in M-SFM with granulocyte–macrophage colony-stimulating factor (GM-CSF, 10 ng/mL, Miltenyi Biotec, Bergisch Gladbach, Germany) for 6 d, as described [46]. Afterwards, macrophages were infected in 6-well plates with single-cell suspensions of *Mtb* at a multiplicity of infection of 5. After 2 h, cells were washed to remove extracellular bacteria and harvested using 1 mM EDTA (Sigma–Aldrich, Steinheim, Germany). Subsequently, 2 × 10^5^ infected macrophages were seeded in 24-well plates and incubated with Gran1 (1 µM) or diluent control for 4 d. To enumerate the number of viable bacilli, infected macrophages were lysed with 0.3% saponin (Sigma–Aldrich, Steinheim, Germany). Cell lysates were re-suspended vigorously, transferred into screw caps, and sonicated in a water bath for 10 min at room temperature. Afterwards, serial dilutions (1:10, 1:100, 1:1000) of sonicates were plated on 7H11 agar plates (BD Biosciences, Franklin lakes, NJ, USA) and incubated for 21 d before determining the number of colonies forming units (CFU).

### 4.14. Statistical Analysis

All statistical analyses, as mentioned in the figure legends, were performed using GraphPad Prism v8.2.1 (GraphPad Software, La Jolla, CA, USA). Significance was calculated using non-parametric tests for paired samples (Wilcoxon rank test, paired *t*-test). Differences were considered significant when *p*-value < 0.05.

### 4.15. Ethical Statement

Zebrafish embryos were used at stages up to 2 days post-fertilization (dpf), which is before they start to feed, at 6 dpf. Embryos that do not yet require feeding are not covered by EU and German animal experiment and welfare legislation (§14 TierSchVersV). Embryos were euthanized at the end of the test by rapid freezing, which is considered the most humane method for euthanasia for fish embryos. Adult fish housing and care was approved by the state of Baden-Württemberg and was monitored by Ulm University animal welfare executives and veterinaries of the city of Ulm.

## Figures and Tables

**Figure 1 ijms-22-08392-f001:**
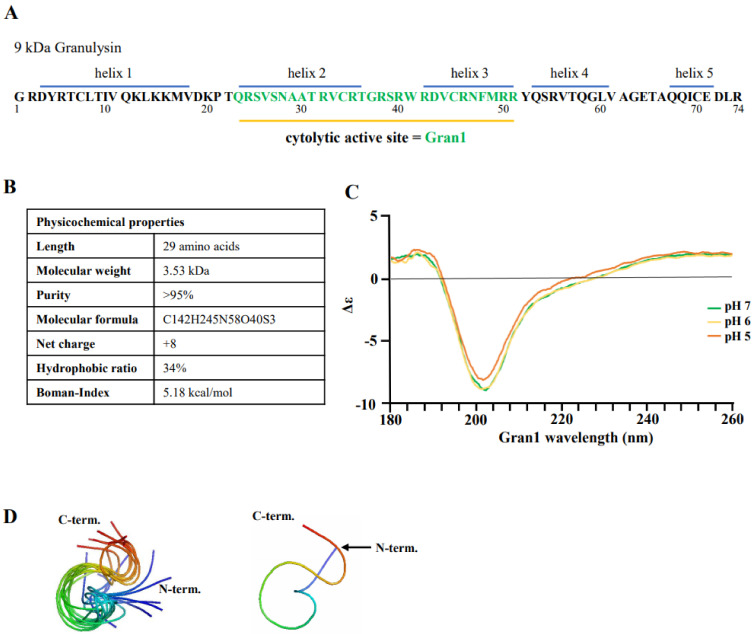
Structure and properties of Gran1: (**A**) Primary sequence and secondary structural elements of Granulysin. The cytolytic active site (amino acids 23–51) of Granulysin was synthesized and named “Gran1”. (**B**) Physicochemical properties of Gran1 were predicted using the Antimicrobial Peptide Databank server APD3 [18] for the molecular weight, amino acid composition, hydrophobic ratio, and Boman index. (**C**) Circular dichroism spectra of synthetic Gran1. The CD spectra were measured at decreasing pH by circular dichroism (*n* = 3). (**D**) Three-dimensional structure of Gran1 was measured. by ^1^H-based nuclear magnetic resonance and calculated with Ambiguous Restraints for Iterative Assignment (ARIA). The Left panel shows the 10 best fitting states for the Gran1 structure. The right panel represents the best fitting state of all ten measurements with the phi and psi angles closest in agreement with the Ramachandran plot.

**Figure 2 ijms-22-08392-f002:**
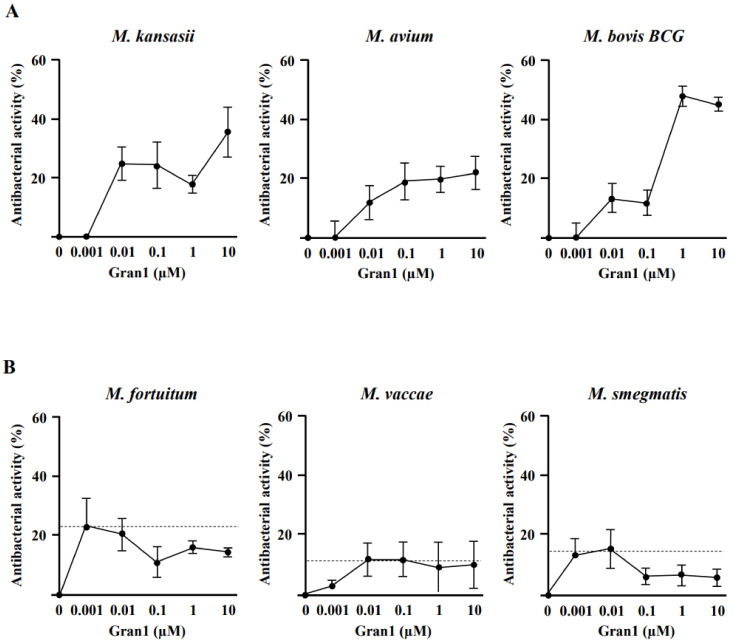
Activity of Gran1 against extracellular mycobacteria: Extracellular mycobacteria (2 × 10^6^) were incubated with increasing concentrations of Gran1 followed by incubation with ^3^H-Uracil for 18 h. The ^3^H-Uracil uptake was measured by scintillation counting. Antibacterial activity was determined for (**A**) slow-growing strains *M. kansasii* (*n* = 4), *M. avium* (*n* = 5), *M. bovis BCG* (*n* = 5) and (**B**) fast-growing strains *M. fortuitum* (*n* = 4), *M. vaccae* (*n* = 5), and *M. smegmatis* (*n* = 5). Data points show the mean antibacterial activity (%) ± SEM calculated from triplicates of each independent experiment. Dotted horizontal lines indicate background activity of diluent control.

**Figure 3 ijms-22-08392-f003:**
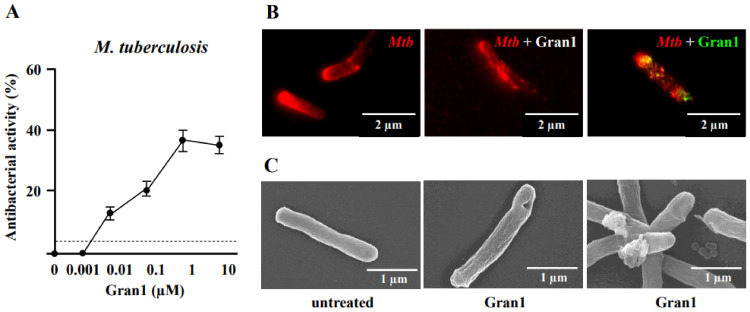
Colocalization of Gran1 with *Mtb* and cell wall distortion: (**A**) Extracellular *Mtb* (2 × 10^6^) were incubated for 72 h with increasing concentrations of Gran1. Afterwards ^3^H-Uracil was added for 18 h, and uptake was measured by scintillation counting. Data points show the mean antibacterial activity (%) ± SEM calculated from triplicates of twenty independent experiments. Dotted horizontal lines indicate background activity of diluent control. (**B**) Extracellular *Mtb* were labelled with NHS-Atto647N and left un-treated (left panel) or were incubated with Gran1 for 30 min (middle and right panel). For signal overlay of mycobacteria and the peptide, Gran1 was labelled using an anti-Gran1 antibody and an Atto594-conjugated secondary antibody (right panel). Samples were acquired by STED microscopy. The middle panel shows Gran-1 treated, NHS-Atto647N-labelled *Mtb*, and the right panel shows an overlay of NHS-Atto647N and Atto594-labelled Gran1. Representative images of minimum ten examined *Mtb* per experiment are shown (*n* = 2). (**C**) Extracellular *Mtb* were treated with Gran1 or left untreated for 3 d and processed for scanning electron microscopy. Representative images show untreated *Mtb* and Gran1-treated *Mtb*. Images were acquired using a Hitachi S-5200 scanning electron microscope. Magnification 40.000× to 50.000×.

**Figure 4 ijms-22-08392-f004:**
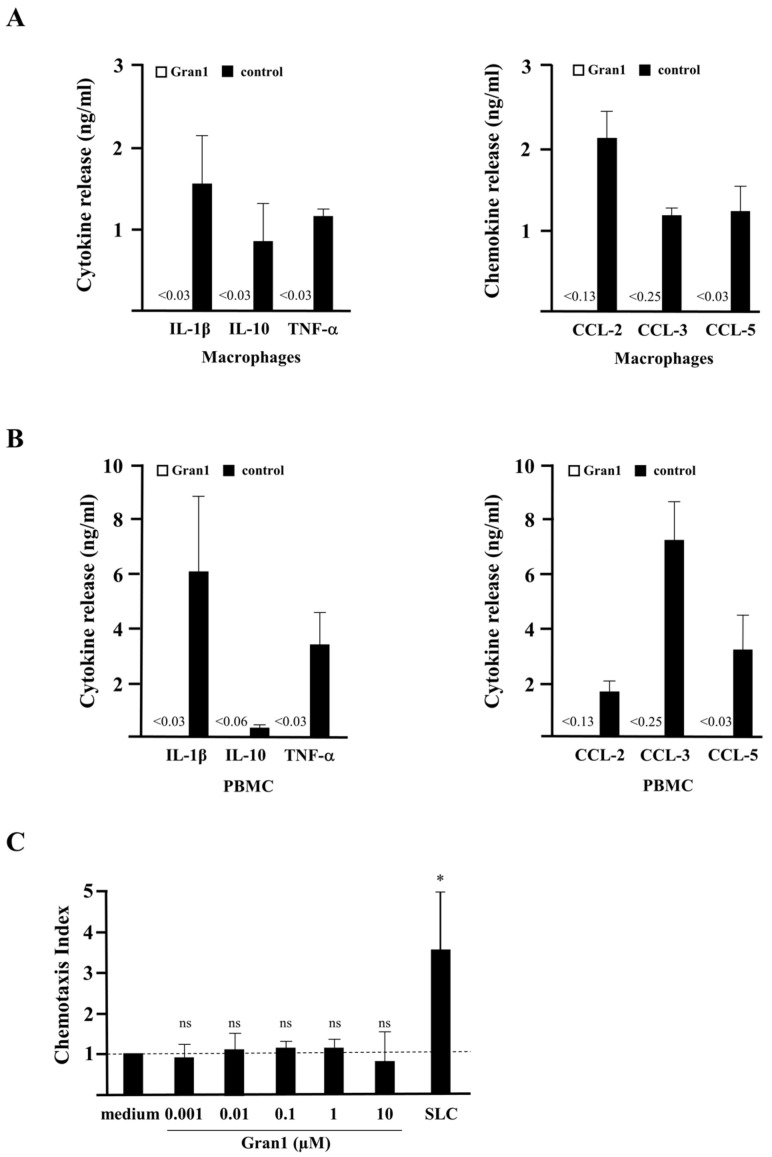
Cytokine release, chemotaxis, and toxicity of Gran1: (**A**,**B**) 0.5 × 10^6^ macrophages or PBMC were incubated overnight with Gran1 or positive controls (LPS, 100 ng/mL or PHA, 20 µg/mL, for CCL-5) and the release of the cytokines (left panels) or chemokines (right panels) was determined after overnight incubation with enzyme-linked immunosorbent assay (ELISA). Background of CCL-2 and CCL-3 signals in unstimulated samples were deducted. The small numbers left from the black bars indicate the sensitivity of each ELISA in ng/mL. Bars represent the mean cytokine release + SEM (*n* = 7). (**C**) PBMC were incubated for 3 h in a chemotaxis chamber in the presence of Gran1 or SLC (5 µg/mL). The chemotaxis index was calculated by comparison with the number of migrated cells in stimulate samples to the number of migrated cells in un-stimulated samples. Bars represent the mean chemotaxis index + SD of five independent donors. Samples were compared to medium control using a paired t-test (SLC: *p* = 0.0240). (**D**) 0.1 × 10^6^ macrophages were incubated with Gran1 for 24 h. Cell viability was measured by PrestoBlue assay. Bars show the mean viability (%) + SEM of five independent experiments. (**E**) Zebrafish embryos were investigated for viability and phenotypes indicative of sublethal toxicity at 48 hpf after exposure for 24 h to Gran1 (1 µM), NRC-03 (positive control causing cytotoxicity), and Aqua ad injectabilia (negative control). The graph shows the affected zebrafish embryos (%) of two independent experiments with *n* = 30 embryos in each group. “Severe damage” indicates lysis or widespread necrosis, which precluded analysis of more specific sublethal phenotypes. “Altered phenotype” includes less severe lysis or necrosis (cytotoxicity), reduced or absent circulation or heart edema (cardiovascular toxicity), developmental delay or malformation (developmental toxicity), and reduced or absent touch response (neurotoxicity).

**Figure 5 ijms-22-08392-f005:**
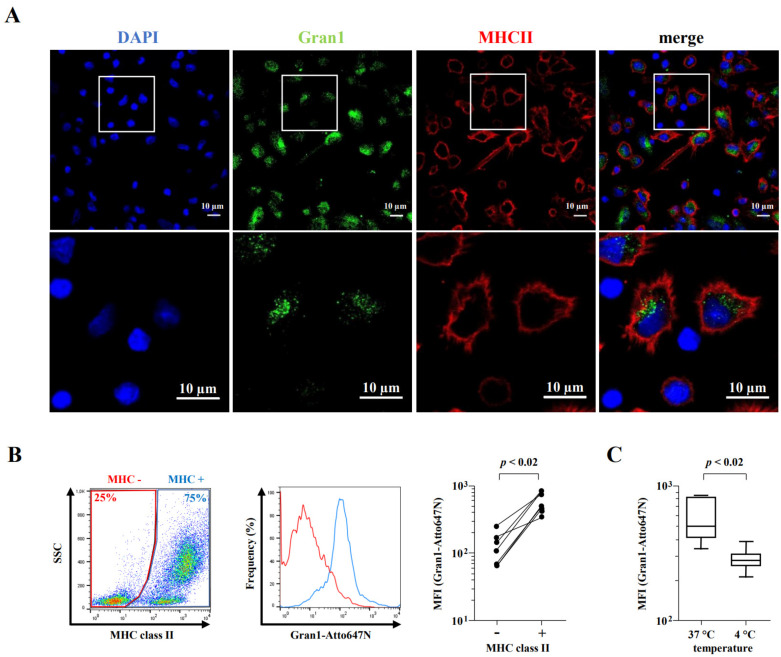
Macrophage-specific uptake of Gran1 in mixed cell culture: (**A**) PBMC mixed with autologous macrophages (ratio 1:1) were incubated with Gran1 for 2 h and labelled using an anti-Gran1 antibody (1:200) and a Cy2-labelled secondary antibody. Cells were stained for MHC class II (red) and cell nuclei were stained with DAPI (blue). Original magnification ×63 (upper four panels) with 3.3 zoomed region of interest (magnification ×210, lower four panels). Data shown are representative for one out of three donors. (**B**) PBMC mixed with autologous macrophages were incubated with Gran1-Atto647N for 2 h, stained for MHC class II (MHCII-FITC), and analyzed by flow cytometry. The left panel shows a representative dot plot (*n* = 7). The histogram shows the percentage of Gran1-positive cells in MHC class II negative (red) and MHC class II positive (blue) population (representative donor, *n* = 7). The results of each individual donor are presented in the right panel. Statistical analysis was performed using a non-parametric Wilcoxon rank test for paired samples (*p* = 0.0156). (**C**) Uptake of Gran1 in MHC class II positive cells after 2 h of incubation at 4 °C and 37 °C. Box plots show the median (horizontal lines) with upper and lower quartile and whiskers indicating minimum and maximum values (*n* = 7). Statistical analysis was performed using a non-parametric Wilcoxon rank test for paired samples (*p* = 0.0156).

**Figure 6 ijms-22-08392-f006:**
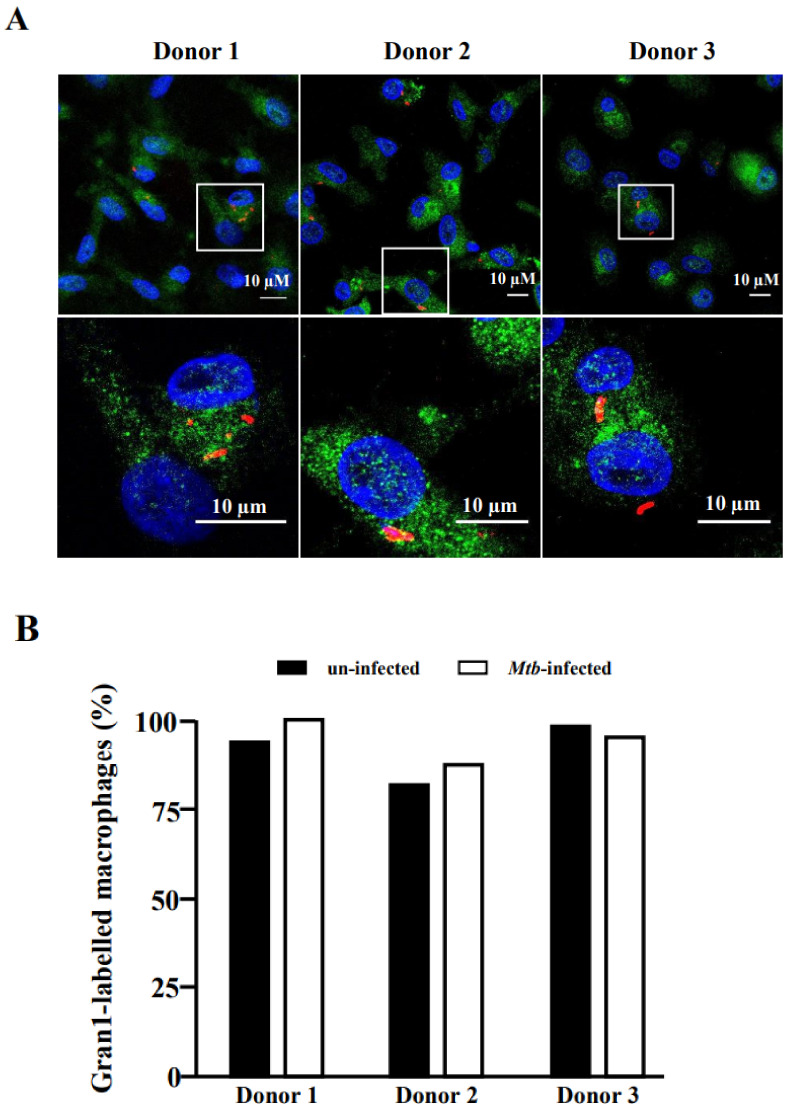
Uptake of Gran1 by *Mtb*-infected macrophages: (**A**) Macrophages were infected with NHS-AlexaFluor 647-labelled *Mtb* for 16 h, followed by 2 h incubation with Gran1. Gran1 was labelled using an anti-Gran1 antibody and a Cy2-labelled secondary antibody (green). Cell nuclei were stained with DAPI (blue). Images were acquired using an inverted laser scanning confocal microscope (Zeiss LSM 710). Images show representative areas for three donors. Original magnification x63 (upper three panels) with a 3.5-fold zoom on the region of interest (lower three panels). (**B**) The graph shows the numberof Gran1-labelled macrophages (%) in un-infected (black) and infected (white) cells for three independent donors.

**Figure 7 ijms-22-08392-f007:**
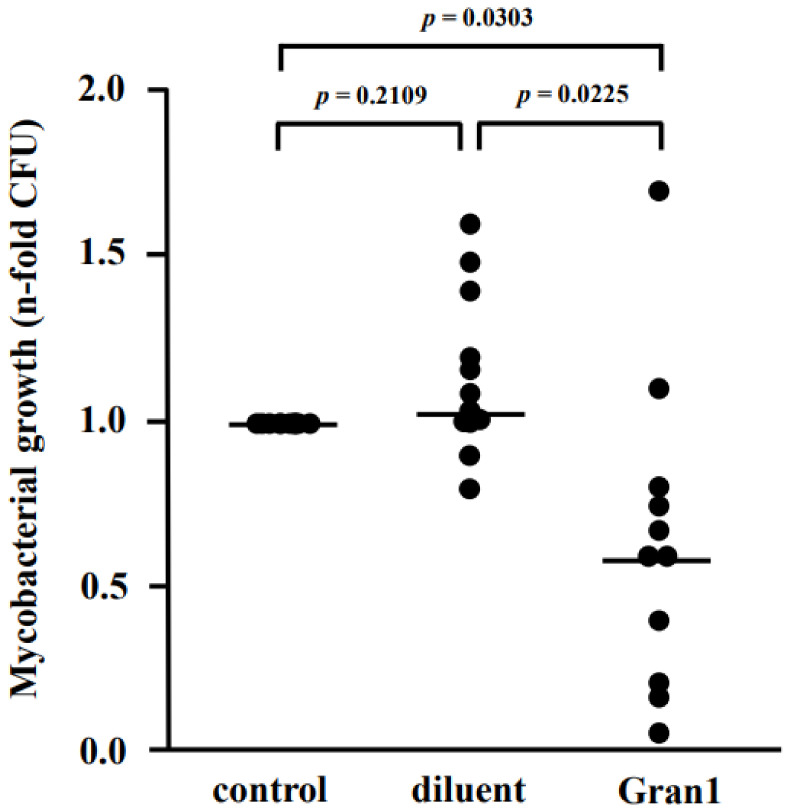
Gran1 inhibits growth of intracellular *Mtb*. Macrophages were infected with *Mtb* followed by incubation with Gran1 for 4 d. Multiplication of *Mtb* was determined by plating cell lysates on 7H11 agar plates. After 21 d of incubation, the number of colony-forming units (CFU) was determined. Lines indicate mean values and dots represent individual values of *n*-fold mycobacterial growth as compared to growth control (d4) for eleven individual donors. Statistical analysis was performed using multiple t-tests (Wilcoxon rank test, *p* = 0.05).

**Table 1 ijms-22-08392-t001:** Mycobacterial strains.

Name	Source
*Mycobacterium tuberculosis*	ATCC 27294 ^a^
*Mycobacterium avium*	ATCC 25291 ^a^
*Mycobacterium kansasii*	ATCC 12478 ^a^
*Mycobacterium bovis BCG*	BCG medac, PZN: 02736484
*Mycobacterium smegmatis*	ATCC 19420 ^a^
*Mycobacterium fortuitum*	ATCC 6841 ^a^
*Mycobacterium vaccae*	ATCC 15483 ^a^

^a^ ATCC (American Type Culture Collection) Manassas, VA, USA.

## Data Availability

The raw data supporting the conclusions of this article will be made available by the authors, without undue reservation, to any qualified researcher.

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
