# Peer review of "Gran1: A Granulysin-Derived Peptide with Potent Activity against Intracellular Mycobacterium tuberculosis"

_ijms, 2021, doi:10.3390/ijms22168392_

Round 1

Reviewer 1 Report

In this uniquely designed and well written experimental study Noschka et al. demonstrate that the putative cytolytic site of the antimicrobial peptide (AMP), Granulysin, (termed “Gran1”) exhibits antimicrobial activity against  Mycobacterium tuberculosis (Mtb) as well as against a panel of clinically relevant non-tuberculous mycobacteria. Gran1 efficiently inhibited the mycobacterial proliferation in the low micro molar range. The authors demonstrate that Gran1 showed no off-target effects  neither in primary human cells (cytokine release, chemotaxis, cell death) nor in zebrafish embryos (cytotoxicity, developmental toxicity, neurotoxicity, cardiotoxicity). Gran1 was selectively internalized by macrophages, the major host cell of Mycobacteria, and homogenously distributed within the cytoplasm which led to restriction of the proliferation of the pathogen. Though the mechanism of how this affects the pathogen within the phagolysosome is still under investigation the paper reveals important insights on the therapeutic potential of Gran1.

Considering the burden of mycobacterial diseases to human health and the limited treatment options the paper is highly relevant.

I have no concerns or suggestions regarding the experimental set up, which is sound. The results are clearly presented and put into the context of the current literature in the discussion. I have only one question: I understand from fig 4 that Gran1 in its "therapeutic" concentration did not induce chemo- or cytokines thus there are no white bars. However, what do the small numbers left from the black bars indicate? Please explain.

I would like to congratulate the authors to their work and highly recommend the manuscript for publication in IJMS.

Author Response

The small numbers left from the black bars indicate the sensitivity of each ELISA in ng/ml. This information is included in the legend of figure 4 (line 185-186). 

Reviewer 2 Report

 In the current manuscript titled “ Gran1: A Granulysin-derived peptide with potent activity against intracellular Mycobacterium tuberculosis”, the author highlighted the anti-mycobacterial activity of Granulysin-derived peptide called Gran1. Gran1 efficiently inhibited the extra- and intracellular Mycobacterial growth in the low micromolar range by interacting with the surface of Mtb, causing lethal distortions of the cell wall. On the other hand, the author also showed that Gran1 did not exhibit a toxic effect on human cells. According to me the data and the experimental design looks very clean, the manuscript is also written well. I have only a few minor suggestions.

  • It would be good to have separate headings for the results section.
  • It would be interesting to look at the anti-bacterial activity of Gran-1 against other pathogenic bacteria.
  • It would be interesting to look at the combinatorial effect of this Gran-1 against Mycobacteria When incubated with frontline anti-TB drugs.

Author Response

It would be good to have separate headings for the results section.

As suggested separate headings for the results section were inserted:

Line 68: Structure and physicochemical characteristics of Gran1

Line 96: Antibacterial activity of Gran1 against extracellular mycobacteria

Line 153: Off-target effects and toxicity of Gran1

Line 197: Internalization of Gran1 by macrophages

Line 238: Antibacterial activity of Gran1 against intracellular Mtb

It would be interesting to look at the anti-bacterial activity of Gran-1 against other pathogenic bacteria. It would be interesting to look at the combinatorial effect of this Gran-1 against Mycobacteria When incubated with frontline anti-TB drugs.

We agree that the next logical step will be to extend the antibacterial spectrum of Gran1. This will  also include the possibility of a synergistic effect of Gran1 with frontline anti-TB drugs. These sophisticated experiments require the establishment of specific antibacterial assays for different clinically relevant bacteria such as Staphylococcus aureus, Klebsiella pneumoniae, Escherichia coli or Pseudomonas aeruginosa. We intend to include the results of these ongoing experiments in a follow up manuscript, which will also present functional studies on Gran1 delivered via nanoparticles such as liposomes and mesoporous nanoparticles. We added this intriguing outlook in the discussion of the revised manuscript (lines 294-296 and lines 359-361).